# Self-Assembled Nanoparticles from Cationic Dipeptides and D-π-A Chromophores for Near-Infrared Photothermal Therapy

**DOI:** 10.3390/ijms262211235

**Published:** 2025-11-20

**Authors:** Wei Zhou, Liangxin Feng, Yanfei Zeng, Jiaxuan Lin, Shuhui Bo, Nan Sun, Xiaoming Zhang

**Affiliations:** 1School of Science, Minzu University of China, Beijing 100081, China; dreamhunter675@163.com (W.Z.); 21010819@muc.edu.cn (L.F.); 17760386611@163.com (Y.Z.); xsjljx@163.com (J.L.); boshuhui@muc.edu.cn (S.B.); 2Optoelectronics Research Centre, Minzu University of China, Beijing 100081, China; 3Hubei Key Laboratory of Novel Reactor and Green Chemical Technology, School of Chemical Engineering and Pharmacy, Wuhan Institute of Technology, Wuhan 430205, China; 4National Engineering Research Center for Colloidal Materials, School of Chemistry and Chemical Engineering, Shandong University, Jinan 250100, China

**Keywords:** cationic dipeptides, supramolecular self-assembly, organic chromophore, near-infrared photothermal therapy

## Abstract

Developing nanoformulations that combine potent photothermal efficacy with robust biocompatibility remains a critical hurdle for precision cancer therapy. Herein, we successfully fabricated CDPNCs-Z3 composite nanoparticles featuring a distinctive spiky architecture via an induced reconstruction self-assembly strategy using cationic dipeptides (CDP). In contrast to simple physical encapsulation, the incorporation of the functional guest molecule Z3 drives the synergistic reconstruction of CDP from fibrous aggregates into smaller, monodisperse particulate nanostructures. This distinct morphological transformation is ascribed to the combined effects of π-π stacking between Z3 and the CDP aromatic system and the presence of strong electron-withdrawing groups. Under 808 nm laser irradiation, these composite nanoparticles demonstrate superior photothermal performance and exceptional cycling stability. In vitro assays further validated their high cellular penetration, negligible dark toxicity, and potent photothermal killing effect. This work not only establishes a versatile new paradigm for building peptide-based nanostructures but also lays a solid foundation for designing safe and effective next-generation photothermal therapeutic agents.

## 1. Introduction

Cancer remains one of the most significant diseases threatening human health, making the innovation and optimization of its therapeutic approaches a central research focus in the biomedical field. In recent years, photothermal therapy (PTT) has emerged as a highly promising strategy for tumor treatment [1,2,3,4,5,6]. This approach utilizes photothermal agents (PTAs) to generate localized hyperthermia under near-infrared (NIR) light irradiation, enabling the efficient and specific ablation of tumor cells. The efficacy of PTT is fundamentally determined by the performance of its PTAs. To date, a variety of PTAs, including inorganic and organic nanomaterials, have been developed [7,8,9]. Among these, organic small-molecule dyes are particularly favored in the photothermal theranostics field due to their excellent biocompatibility, facile metabolic clearance, and absence of long-term toxicity [10,11,12,13,14,15]. Specifically, organic molecules that exhibit a photothermal conversion effect within the NIR window—which offers superior tissue penetration—represent the forefront of research in cancer diagnosis and therapy [16,17,18,19,20]. However, most currently available NIR-absorbing organic dyes still face some formidable challenges as bioavailable agents, such as the hydrophobicity and short circulation half-life.

To overcome the inherent limitations of individual organic small-molecule dyes, researchers have begun to explore supramolecular self-assembly strategies. By integrating functional molecules with highly biocompatible building blocks, it is possible to construct novel nanophotothermal agents with synergistically enhanced properties [21,22,23,24]. Peptides, as an important class of self-assembling biomolecules, have been confirmed as ideal carriers for constructing supramolecular photothermal nanomedicines, owing to their structural designability, good biodegradability, and low immunogenicity [25,26,27,28]. The diphenylalanine (FF) motif, the primary recognition sequence for the amyloid fibril formation of the Alzheimer’s Aβ-polypeptide [29], and its derivatives have been widely used in the biomedical field. As building blocks, they assemble into various nanostructures—such as nanotubes, spherical vesicles, nanofibers, and nanowires—for applications including drug delivery, tissue engineering, and biosensing [30,31,32]. By modifying the terminal -COOH group of FF to -NH_2_, the resulting cationic diphenylalanine dipeptide (CDP) can aggregate into nanotubes at physiological pH. Upon dilution, these cationic dipeptide nanotubes spontaneously transform into vesicles, enabling the transmembrane delivery of oligonucleotides into HeLa cells [33]. Furthermore, a CDPNP/hpDNA nanofluorescent probe, designed leveraging the charge specificity of CDP, has been shown to overcome the issue of false-negative signals, as the optical interaction between the CDP nanoparticle and the fluorophore is minimal and does not interfere with the fluorescence signal [34].

Herein, we report the development of a novel supramolecular photothermal nanocarrier, designated CDPNCs-Z3, through a facile self-assembly of a biocompatible CDP and a near-infrared chromophore, FTC-3f (synthesized previously in our group, hereafter Z3) [35]. The chromophore Z3 is based on a donor-π-acceptor (D-π-A) framework, which endows it with strong and broad absorption within the 600–850 nm NIR window, with a maximum absorption peak at 710 nm. The as-prepared assembly was subjected to systematic characterization of its morphology, size, and optical properties. Subsequently, its key performance metrics—including Z3 encapsulation efficiency, photothermal conversion capability, and in vivo antitumor activity—were comprehensively assessed. This research seeks to effectively enhance the photostability and photothermal conversion efficiency of organic dyes through a peptide-based supramolecular assembly strategy, thereby providing new design concepts and experimental evidence for the development of safe and efficient novel photothermal theranostic agents for cancer.

## 2. Results

Uniform nanoparticles were fabricated by exploiting the concentration-dependent self-assembly of CDP, which forms a fibrous gel at high concentrations and reorganizes into vesicles upon dilution [36]. The synthesis involved crosslinking peptide CDP with Z3 in the presence of glutaraldehyde (GA) (Figure 1). The morphological consequences of Z3 incorporation were investigated via scanning electron microscopy (SEM) and transmission electron microscopy (TEM).

Prior to dilution, the nanofibrous gel architecture was maintained upon Z3 addition, albeit with enhanced fiber flexibility and surface roughness (Figure 2a,b). Subsequent dilution induced the formation of spherical nanoparticles in both formulations (Figure 2c,d). Notably, the Z3-containing assemblies exhibited a significantly smaller particle size. This size reduction was quantitatively confirmed by dynamic light scattering (DLS), revealing a stark contrast in average diameters: 540 nm for CDPNCs versus 136 nm for the CDPNCs-Z3 complexes. This nearly four-fold decrease is further supported by particle size distribution histograms (Figure 2e and Figure 2f, respectively) derived from SEM images, which capture the particles’ true physical dimensions. The agreement between these two independent methods validates that the size reduction is a systematic and inherent property of the Z3-induced assembly.

Further morphological analysis showed that the pristine CDPNCs displayed relatively large sizes and smooth surfaces (Figure 3a,b). In contrast, assembly with Z3 molecules resulted in a significant size reduction and a transition from smooth surfaces to uniformly distributed, spiky-like nanostructures in the CDPNCs-Z3 composites (Figure 3c–f). The characteristic spiky morphology was observed consistently across all experimental repetitions, which confirms the high reliability of the self-assembly process.

The size-tuning effect of the Z3 feed ratio on the resulting assemblies was systematically explored. As depicted in Figure 4, increasing the Z3 concentration progressively reduced the particle size of the CDPNCs-Z3 assemblies. At a CDP: Z3 mass ratio of 2:1 (Figure 4a), the system displayed a heterogeneous morphology. Raising the ratio to 1:1 (Figure 4b) yielded significantly smaller particles (250–300 nm) with a more homogeneous distribution. The optimal ratio of 1:2 (Figure 4c) produced stable, monodisperse nanoparticles averaging 130–140 nm, a finding in excellent agreement with DLS measurements and SEM histogram in Figure 2f. At an excessive Z3 ratio of 1:3 (Figure 4d), the nanoparticles formed upon dilution were highly prone to aggregation. Pure Z3 was unable to self-assemble into particles (Appendix A). Based on the evaluation of particle size and stability (Appendix A), the 1:2 mass ratio was identified as the optimal formulation.

To elucidate the driving forces behind the assembly, UV-Vis and infrared (IR) spectroscopy were performed. The UV-Vis spectrum of the CDPNCs-Z3 composite retains the characteristic absorption peak of Z3 at ~710 nm. As shown in Figure 5a, a weak shoulder appears at a higher energy relative to the monomer absorption in the CDPNCs-Z3 system. This spectral change suggests the presence of some intermolecular interactions or a specific aggregation state, but its exact nature requires further investigation. The enhanced band intensity and the appearance of shoulder peaks suggest strong intermolecular coupling, likely involving π-π stacking interactions, which can facilitate non-radiative decay pathways, a key factor for high photothermal conversion efficiency [37]. IR analysis (Figure 5b) showed that, compared to pristine CDP molecules, CDPNCs display intensified peaks at 2932 cm^−1^ and 1435 cm^−1^ [38]. Upon complexation with Z3, these peaks shift to 2961 cm^−1^ and 1467 cm^−1^, respectively. An additional peak at 2929 cm^−1^, corresponding to the C-H stretch of Z3 itself [39], was also observed. A new shoulder peak emerged at 1723 cm^−1^ in the composite. In the aromatic region (1450–1600 cm^−1^), the aromatic C=C stretching peak in the CDP-Z3 composite shifts to a lower frequency. Finally, the intensity of the peak at 1268 cm^−1^, characteristic of the C-F bond in the –CF3 group, markedly increased.

The photothermal performance of the CDPNCs-Z3 assembly was assessed using an infrared thermal camera under 808 nm laser irradiation. At a power density of 1.0 W/cm^2^, the solution temperature peaked at 60 °C (Figure 6a). A clear positive correlation between concentration and heating efficiency was observed, with the 40 μg/mL sample reaching ~60 °C after 400 s of irradiation (Figure 6b). The assembly exhibited exceptional photothermal stability over five consecutive laser on-off cycles, with negligible performance degradation (Figure 6c), which demonstrates its robust structural integrity under repeated photothermal stress. The photothermal conversion efficiency (PCE) of the assembly was calculated to be 29.8% at 808 nm (Figure 6d).

For intracellular studies, the internalization of the composite nanoparticles in Hela cells was investigated. The nanoparticles, visualized by the red fluorescence of their Schiff base [40], were rapidly internalized via endocytosis within 1–2 h (Figure 7a). An MTT assay revealed minimal dark toxicity, with cell viability staying above 80% at concentrations up to 90 μg/mL (Figure 7b). This result serves as indirect evidence that any residual glutaraldehyde is negligible and does not impact cell viability. Under 808 nm laser irradiation, the nanoparticles exhibited dose-dependent photothermal cytotoxicity. Cell viability plummeted to approximately 40% at a concentration of 20 μg/mL and was further suppressed to around 30% at 50 μg/mL (Figure 7c).

## 3. Discussion

The concentration-dependent self-assembly of CDP provided a robust foundation for nanoparticle formation. A striking finding was the dramatic morphological evolution upon Z3 incorporation, where the initial large, smooth CDPNCs transformed into smaller, spiky nanostructures. This unique morphology is not merely a structural curiosity, but also offers significant functional advantages. Compared to conventional smooth nanospheres, the high surface-area-to-volume ratio of the spiky architecture is expected to enhance cellular membrane interaction and internalization efficiency [41]. This observation reveals a key self-reconstructive property of CDP, where its interaction with a functional guest molecule actively drives the formation of a more advanced nanostructure, creating an ideal platform for effective guest molecule loading and delivery.

The formation of spiky or anisotropic nanostructures to enhance functional performance is a well-established strategy in nanomedicine. It is widely reported that gold nanostars, with their sharp protrusions, exhibit substantially higher photothermal conversion efficiency and cellular uptake compared to their spherical counterparts [42]. This superior performance is attributed to the “lightning-rod effect,” where electromagnetic fields are massively amplified at the sharp tips, leading to efficient light-to-heat conversion [42]. Furthermore, theoretical studies have suggested that non-spherical geometries, such as rods or spiky particles, promote more efficient interaction with the cell membrane, thereby facilitating receptor-mediated endocytosis [43]. While our CDPNCs-Z3 system is based on an organic self-assembly platform, it leverages the same fundamental principle. The spiky morphology we observed is therefore not merely a structural curiosity, but a functional feature designed to mimic the advantages of inorganic nanostars, potentially leading to enhanced photothermal ablation of tumors and improved intracellular delivery.

The systematic investigation of the Z3 feed ratio demonstrated a clear size-tuning effect, with the 1:2 mass ratio identified as optimal for producing stable, monodisperse nanoparticles. This phenomenon is likely governed by a dynamic equilibrium involving a competition between π-π stacking and steric hindrance. At a low Z3 concentration (CDP:Z3 = 2:1), Z3 can only partially reconstruct the CDP particles, leading to contracted regions where binding occurs, while unbound CDP domains remain in their original state. As the Z3 concentration increases, more Z3 molecules interact with CDP, pulling and tightening its structure, which is the fundamental reason for the size reduction. A qualitative shift occurs at the 1:2 ratio. At this point, the π-π stacking between the benzothiophene core of Z3 and the aromatic system of CDP likely transitions from a simple face-to-face manner to an offset stacking mode. This mode is more conducive to helix formation, thereby guiding the transition of CDP from random fibers or sheet-like structures into more ordered helical structures. Concurrently, this optimal ratio ensures that Z3 is not only sufficient to complete the reconstruction process but also forms a saturated protective layer on the newly formed nanoparticles. This layer is crucial for ensuring stable monodispersity. However, when Z3 is in excess (CDP:Z3 = 1:3), this surface stability is compromised, leading to particle aggregation. The aggregation observed at a higher Z3 ratio (1:3) and the inability of pure Z3 to self-assemble underscore the essential role of CDP as the structural backbone, providing the necessary framework for stable nanoparticle formation.

The driving force behind this morphological evolution appears to be the strong π-π stacking interactions between the planar Z3 molecules. This is consistent with the work of Yang et al. [44], who reported that aromatic guest molecules could reorganize peptide-based fibers into spherical micelles. However, our system goes a step further. Instead of forming simple spheres, the synergistic interaction between Z3 and the cationic dipeptide (CDP) directs the assembly into a more complex, spiky architecture. This suggests that the CDP is not a passive host but an active participant whose final structure is programmable by the nature of the guest molecule.

This hypothesis is further supported by the UV-Vis spectroscopy data. In the assembled state, the absorption bands broaden and a shoulder appears at higher energy relative to the monomeric spectrum in DMSO, indicating significant changes in the electronic environment of the chromophores [45]. This confirms that the Z3 molecules are not simply encapsulated but are densely packed and ordered within the nanostructure, a direct consequence of the self-reconstructive assembly process. This tight packing is often correlated with enhanced photostability and photothermal conversion, providing a molecular-level explanation for the excellent performance of our platform.

The IR spectral shifts offer further mechanistic insights. Electron-withdrawing groups typically cause a shift to higher wavenumbers, while electron-donating groups lead to a shift to lower wavenumbers. The observed shifts to higher wavenumber of the C-H and C=N stretching bands in the FTIR spectra are attributed to the inductive effect of the strongly electron-withdrawing –CF3 group in Z3. The emergence of a new shoulder peak at 1723 cm^−1^ suggests that electron-withdrawing groups in Z3, such as the ester (-COO), pull electron density away from the C=O bond, strengthening it. The counterintuitive lower frequency shift of the aromatic C=C stretching peak is likely caused by π-π stacking between Z3’s benzothiophene core and CDP’s aromatic system, which perturbs the electron distribution. The marked increase in the C-F bond peak intensity serves as definitive proof of the successful incorporation of Z3.

The CDPNCs-Z3 assembly demonstrated rapid and efficient photothermal heating, achieving temperatures well above the 42–45 °C threshold required for effective tumor ablation [46]. This highlights its considerable promise as a photothermal therapeutic agent. While the calculated photothermal conversion efficiency (PCE) of 29.8% for the Z3 nanoparticles at 808 nm is highly promising, it is lower than the ~51% reported for the free molecule [35]. We attribute this difference to the photophysical consequences of nano-confinement and aggregation, which restrict intramolecular motions. These effects alter the non-radiative decay pathways, potentially opening less efficient channels or causing aggregation-caused quenching that impedes heat generation.

To further contextualize the performance of our CDPNCs-Z3 NPs, it is essential to compare them with established benchmark photothermal agents. Indocyanine green (ICG), a FDA-approved NIR dye, is a natural point of reference. While ICG is clinically used, its application is significantly hampered by poor photostability, rapid clearance from the body, and a tendency to aggregate in aqueous environments, all of which compromise its therapeutic efficacy and reproducibility [47]. In stark contrast, our CDPNCs-Z3 NPs demonstrate excellent aqueous stability and photostability under continuous irradiation, addressing the key limitations of ICG and offering a more robust platform for repeated or prolonged treatments.

When compared to high-performance inorganic nanoagents, such as platinum-based nanoparticles, the picture becomes more nuanced. Certain Pt nanostructures have reported PCEs exceeding 60% [48], surpassing the 29.8% of our CDPNCs-Z3 NPs. We acknowledge this superior intrinsic heating efficiency. However, this metric alone does not dictate clinical success. The translation of inorganic agents like Pt is often complicated by concerns over long-term biocompatibility and potential heavy-metal toxicity [49]. Furthermore, their synthesis can be costly and complex. Our CDPNCs-Z3 NPs, being composed of organic molecules, present a more favorable biocompatibility profile and a potentially simpler, more scalable synthesis. Therefore, the observed PCE of 29.8% for CDPNCs-Z3 NPs represents a highly promising and pragmatic trade-off, balancing potent photothermal activity with the essential attributes of stability, biocompatibility, and translational feasibility that are paramount for in vivo applications.

It is crucial to note, however, that the choice of an 808 nm excitation source is a deliberate and necessary strategy for in vivo applications. This wavelength resides within the first near-infrared (NIR-I) biological window, providing superior tissue penetration and minimal background interference—features that are paramount for treating deep-seated tumors. While this introduces a spectral mismatch with the assembly’s 710 nm absorption peak, the resulting PCE remains highly promising. More importantly, the nanoparticle formulation endows Z3 with essential clinical translational properties, including aqueous stability, biocompatibility, and passive tumor targeting via the EPR effect, which the free molecule lacks. Consequently, the observed performance represents a highly acceptable trade-off, balancing a slight reduction in photophysical efficiency with the profound practical advantages required for a robust and effective photothermal agent.

The in vitro studies confirmed the suitability of the composite nanoparticles for biomedical applications. The rapid internalization by Hela cells and high biocompatibility, evidenced by cell viability remaining above 80% at high concentrations, establish a solid foundation for intracellular therapy. Crucially, the nanoparticles exhibited potent, dose-dependent photothermal therapeutic effects under laser irradiation. Collectively, these findings establish the CDPNCs-Z3 composite nanoparticles as an ideal phototherapeutic agent, characterized by low dark toxicity and high phototoxicity, which highlights their significant potential for targeted tumor therapy. Future work will focus on validating these promising results across a broader panel of cancer cell lines and in vivo assessment to establish the generalizability of the therapeutic effect. A key direction will be the integration of these experimental data with computational modeling, an approach that holds the promise of revealing the complex regulatory mechanisms at play.

## 4. Materials and Methods

### 4.1. Preparation of CDPNCs-Z3 Assemblies

First, 1 mg of CDP (GL Biochem Ltd., Shanghai, China) was dissolved in 10 μL of hexafluoroisopropanol (HFP, Sigma-Aldrich, Shanghai, China). Then, a specified volume of the Z3 stock solution (e.g., 50 μL of a 10 mg/mL solution in HFP) was added, followed by the addition of Opti-MEM medium (Biosharp, Shanghai, China) to bring the total solution volume to 100 μL. While the solution was being sonicated, 900 μL of a 0.06% aqueous glutaraldehyde (GA, Sigma-Aldrich, Shanghai, China) solution was added (as depicted in Figure 1). The mixture was then allowed to age at room temperature for 24 h, until a dark green precipitate formed at the bottom of the centrifuge tube. Subsequently, the precipitate was then collected by centrifugation. Purification was performed by repeated washing with deionized water until the supernatant was nearly colorless, confirming the removal of unreacted glutaraldehyde. Finally, the washed precipitate was redispersed in deionized water and stored at 4 °C. To ensure high reproducibility, all experiments were conducted by strictly adhering to the described protocol.

### 4.2. Morphological Characterization of CDPNCs-Z3

#### 4.2.1. Scanning Electron Microscopy (SEM) Analysis

The surface morphology of the CDPNCs-Z3 assemblies was characterized using an S-4800 scanning electron microscope (SEM). The sample was drop-cast onto a cleaned silicon wafer and allowed to dry at room temperature (25–30 °C). The dried wafer was then fixed onto the sample stage using conductive tape and subjected to gold sputtering with parameters set at I = 30 mA and T = 300 s. After gold coating, the sample stage was inserted into the SEM chamber (operating at 15 kV) for imaging and observation.

#### 4.2.2. Transmission Electron Microscopy (TEM) Analysis

The CDPNCs-Z3 assemblies were further characterized using a JEM-1011 transmission electron microscope (TEM). A small aliquot of the sample was pipetted onto a copper grid and allowed to dry at room temperature before being inserted into the TEM. TEM images were recorded at an acceleration voltage of 100 kV.

#### 4.2.3. Particle Size Analysis

The particle size of CDPNCs-Z3 was analyzed using a Malvern laser particle size analyzer. The CDPNCs-Z3 were ultrasonically dispersed in deionized water, and 1 mL of the dispersed sample was added to the sample cell. The cell was then placed in the laser particle size analyzer for testing, and the data were recorded.

### 4.3. Spectroscopic Characterization

#### 4.3.1. Ultraviolet-Visible (UV-Vis) Absorption Spectroscopy

The UV-Vis absorption spectrum of the CDPNCs-Z3 assemblies was measured using a Hitachi U-3900 UV-Vis spectrophotometer. The purified assembly particles were first dried at 30 °C and then redispersed in a cuvette containing 2.5 mL of DMSO (Sigma-Aldrich, Shanghai, China). The spectral scan was performed over a wavelength range of 500–1000 nm with a slit width of 1 nm at room temperature. DMSO was used as the baseline for the measurement.

#### 4.3.2. Fourier-Transform Infrared (FTIR) Spectroscopy

FTIR analysis was conducted using a Thermo Fisher Scientific FTIR IS5 spectrometer. The nanoparticles were dried at room temperature. The dried sample was then mixed with potassium bromide (KBr) at a mass ratio of 1:200. The mixture was ground thoroughly in a mortar to ensure homogeneity, subsequently pressed into a pellet, and placed in the instrument for scanning. While the pressure applied during pellet formation can minimally affect band positions, control experiments with a non-interacting standard suggest this effect is negligible under our conditions.

### 4.4. Cytotoxicity Assay

HeLa cells were maintained in Dulbecco’s Modified Eagle Medium (DMEM, Biosharp, Shanghai, China) containing 10% fetal bovine serum (FBS, Noverse, China) and 1% penicillin-streptomycin (50 mg/mL each, Beyotime, Shanghai, China) at 37 °C in a humidified atmosphere of 5% CO_2_. For cytotoxicity evaluation, Hela cells (China Center for Type Culture Collection) were seeded into 96-well plates (8000 cells/well) and exposed to a concentration gradient of the material (0–90 μg/mL) for 24 h. For the phototoxicity assessment, following a 4-h incubation, cells were washed with PBS, and the medium was replenished with fresh DMEM. Subsequently, the cells were subjected to 10 min of NIR irradiation, followed by a further 24-h incubation. The viability of HeLa cells in all groups was determined by a standard MTT assay.

### 4.5. Photothermal Performance Characterization of CDPNCs-Z3 Assemblies

To evaluate the photothermal effect, 200 μL of CDPNCs-Z3 suspensions at various concentrations were added to the wells of a 96-well plate. Deionized water and a suspension of CDPNCs (without Z3) were used as the control groups. The samples were then irradiated with an 808 nm laser at a power density of 1.0 W/cm^2^ for 6 min under ambient conditions (25 °C). During irradiation, the temperature of each solution was monitored and recorded every 30 s using an infrared (IR) thermal camera.

Furthermore, to assess the photostability, the CDPNCs-Z3 suspension was subjected to five laser on/off cycles. Each cycle consisted of 6 min of laser irradiation followed by 10 min of natural cooling.

The photothermal conversion efficiency (η) of CDPNCs-Z3 was calculated using the following equation:$$\eta =\frac{hA(\Delta T_{max,min}-\Delta T_{max,H_{2}O})}{I(1-10^{{-A}_{\lambda }})}\times 100\%$$

Here, $∆T_{max,min}$ is the maximum temperature change in the assembly suspension, and $∆T_{max,H_{2}O}$ is the maximum temperature change in the pure water control. *I* represents the power density of the 808 nm laser, and $A_{\lambda }$ is the absorbance of the assembly at the excitation wavelength (808 nm). The value of *hA* is determined from the linear fit of the cooling temperature data, according to the following relationship:$$\theta =\Delta T_{time}/\Delta T_{max,min}$$$$t=\frac{\sum_{i}m_{i}C_{\rho ,i}}{hA}ln\theta$$

Here, $\Delta T_{time}$ is the temperature change at each measured time interval during the cooling process. The mass of the suspension (*m_i_*) is 0.2 g, and the specific heat capacity of the solvent (*C_ρ,i_*) is 4.18 J/(g·°C).

## 5. Conclusions

In this study, we report a novel “induced-reconstruction self-assembly” strategy to fabricate a superior photothermal nanoplatform. The key finding is that the functional chromophore Z3 acts not as a passive cargo, but as an active “structural driver” that reconstructs CDP fibers into uniform, spiky nanoparticles. This mechanism, synergistically driven by host-guest π-π stacking and electron-withdrawing effects, provides a new paradigm for the controllable fabrication of intelligent nanostructures with advanced therapeutic functions.

Leveraging this strategy, the fabricated CDP/Z3 composite nanoparticles demonstrate remarkable advantages in key performance metrics. Their 136 nm size and distinctive surface morphology confer highly efficient cellular internalization. Furthermore, they achieve an exceptional combination of a high photothermal conversion efficiency (29.8%) and excellent biocompatibility (low dark toxicity), ensuring both high efficacy and safety for tumor therapy.

The significance of this work extends beyond the development of a superior photothermal agent. It establishes a versatile “induced-reconstruction self-assembly” strategy, paving a new pathway for the design of functional, intelligent, short peptide-based nanomaterials. This innovative strategy can be broadly adapted for the co-assembly of various drug molecules, diagnostic probes, and peptide scaffolds, thereby providing powerful theoretical guidance and technical support for the design and development of the next generation of highly efficient and precise nanomedicines. Future work will focus on synthesizing or selecting D-π-A structure molecules with red-shifted absorption. Leveraging the preparation paradigm from this study, we aim to develop photothermal agents with high conversion efficiency and superior biocompatibility, offering new strategies and a material foundation for deeper, more effective tumor phototherapy.

## Figures and Tables

**Figure 1 ijms-26-11235-f001:**
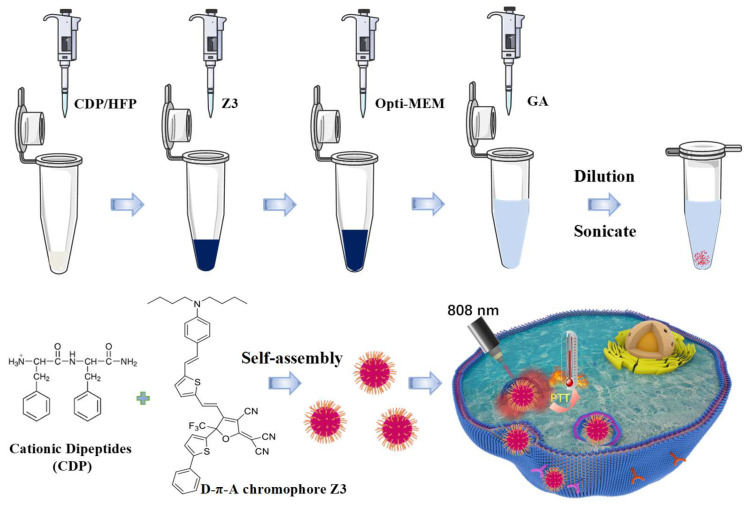
Schematic illustration of the assembly of CDPNCs-Z3 composite nanostructures.

**Figure 2 ijms-26-11235-f002:**
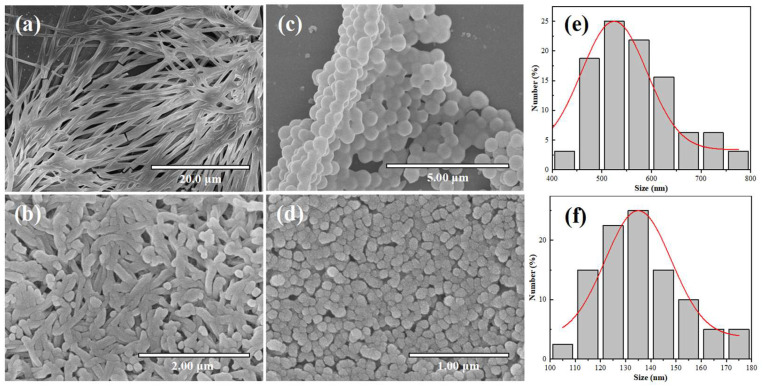
Formation process of CDPNCs particles and CDPNCs-Z3 composite particles. (**a**,**b**) SEM images of the fibrous precursor structures formed at high concentration. (**c**,**d**) Spherical particulate structures formed after dilution. (**e**,**f**) Particle size distribution histograms obtained from the SEM images.

**Figure 3 ijms-26-11235-f003:**
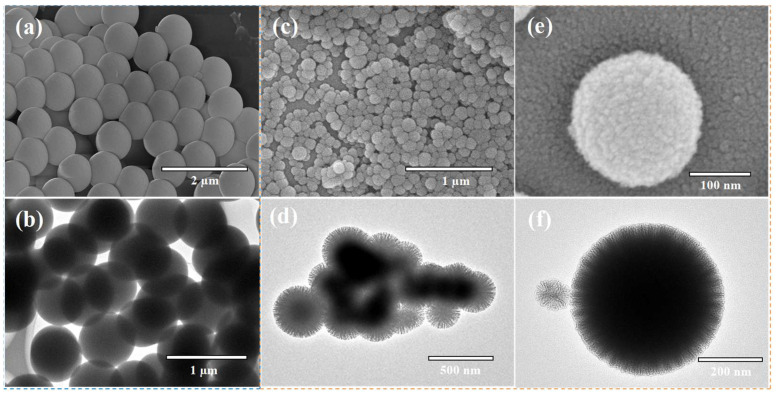
Morphological and structural characterization of CDPNCs and CDPNCs-Z3 composite nanoparticles. (**a**,**b**) SEM (**a**) and TEM (**b**) images of the CDPNCs. (**c**,**d**) SEM (**c**) and TEM (**d**) images of the CDPNCs-Z3 composite nanoparticles, showing their overall morphology. (**e**,**f**) High-magnification SEM (**e**) and TEM (**f**) images of a single CDPNCs-Z3 particle, revealing its surface structure in detail.

**Figure 4 ijms-26-11235-f004:**
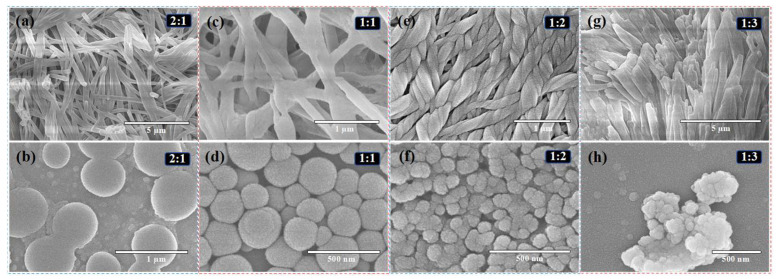
Effect of CDP/Z3 mass ratio and concentration on the self-assembled morphology. SEM images of CDPNCs-Z3 composites with CDP/Z3 mass ratios of (**a**,**b**) 2:1, (**c**,**d**) 1:1, (**e**,**f**) 1:2, and (**g**,**h**) 1:3, respectively. The top row (**a**,**c**,**e**,**g**) shows the structures at high-concentration conditions, and the bottom row shows the structures after dilution.

**Figure 5 ijms-26-11235-f005:**
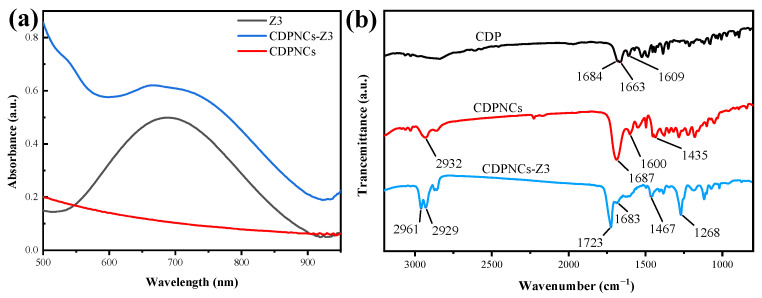
(**a**) UV-Vis absorption spectra of Z3, CDPNCs and their components. Measurements were performed in DMSO at a concentration of 20 μM and room temperature. (**b**) Fourier-transform infrared (FTIR) spectra of the CDPNCs-Z3 composite nanoparticles and their components.

**Figure 6 ijms-26-11235-f006:**
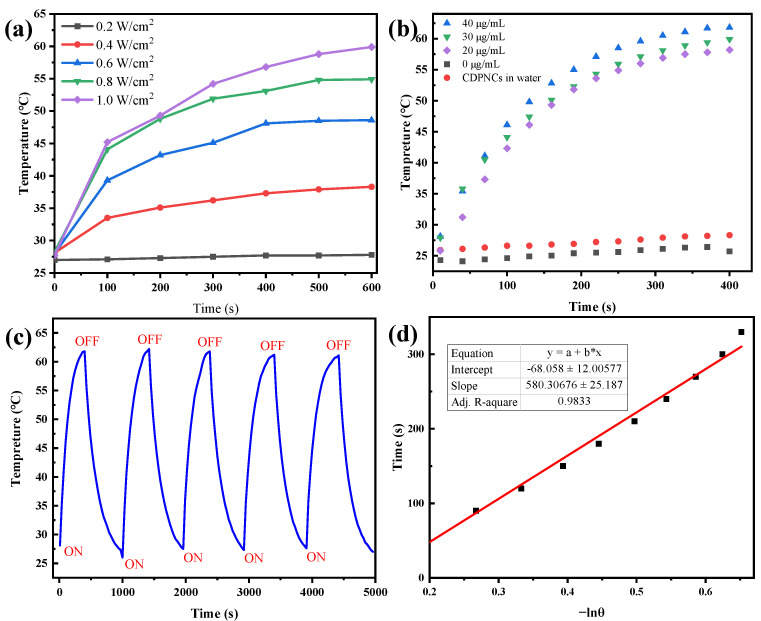
Evaluation of the photothermal performance of the CDPNCs-Z3 composite nanoparticles. (**a**) Temperature elevation as a function of irradiation time under different laser powers. (**b**) Photothermal heating curves at various concentrations. (**c**) Photothermal cycling stability test. (**d**) Evaluation of the photothermal conversion efficiency.

**Figure 7 ijms-26-11235-f007:**
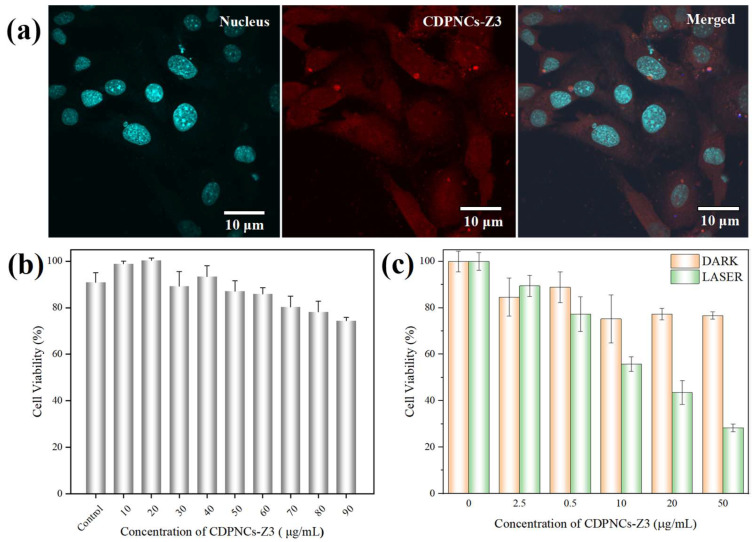
Evaluation of cellular uptake and in vitro photothermal therapeutic effects of the CDPNCs-Z3 composite nanoparticles. The nuclei were stained with Hoechst 33342. (**a**) Distribution of CDPNCs-Z3 in HeLa cells. (**b**) Evaluation of the dark cytotoxicity of the composite. (**c**) Photothermal-induced cell killing efficacy.

## Data Availability

The original contributions presented in this study are included in the article. Further inquiries can be directed to the corresponding author.

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
