# Peer review of "Self-Assembled Nanoparticles from Cationic Dipeptides and D-π-A Chromophores for Near-Infrared Photothermal Therapy"

_ijms, 2025, doi:10.3390/ijms262211235_

Round 1

Reviewer 1 Report

Comments and Suggestions for Authors

Dear Authors.

The manuscript number ijms-3966673 is accepted with major revision. It is well written. However, the authors need to explain the following points before the final acceptance.

The FTIR measurements presented here were obtained using a mortar and pressed into a pellet. Is there any pressure effect on the shifting band in the Infrared spectra?  When weak interactions are involved, the point takes relevance. Could you explain and add some comments to the manuscript, please?

In the discussion section, the blue-shifts of the C-H and C=N peaks attributed to the inductive effect of the –CF3 group are related to Infrared or electronic absorption measurements. Could you clarify it, please?

Figure 6d needs to be improved. The slope has errors, and the number of significant figures. Could you verify it.

The term “A blue-shift” in the text, as used in the context of the infrared bands at 2961 cm-1 and 1467 cm-1, is confusing because it is more associated with UV-Vis spectra than with infrared. Should explain better this term. For example, this shift is more related to low or high wavenumbers, which are also associated with withdrawing or donor effect groups. Could you verify the text and make any changes.

The histograms present show a kurtosis degree that is not explained in the text. Also, Figure 4 is indeed supramolecular behavior; however, it is not explained in the text. Why is there a difference in the elongated fibers between the 2:1 and 1:2 molar ratios? In the last one, the thread fibers seem denser. Could the authors explain this point, which should affect the photothermal properties.

Reviewer 2 Report

Comments and Suggestions for Authors

The manuscript titled “Self-Assembled Nanoparticles from Cationic Dipeptides and D-π-A Chromophores for Near-Infrared Photothermal Therapy,” which begins with the abstract: “Developing nanoformulations that combine potent photothermal efficacy...”, has been reviewed. The manuscript is generally well-written and presents a novel approach with promising results. However, several important issues need to be addressed before it can be considered for acceptance. Below are my comments:  

Recommendation: Major Revisions: may be suitable for publication, but only after substantial changes

  • The concept of “induced-reconstruction self-assembly” is novel and well-articulated, but the manuscript would benefit from a clearer mechanistic explanation supported by additional experimental evidence (e.g., molecular dynamics or computational modeling).
  • The photothermal conversion efficiency (PCE) of 29.8% is promising; however, the discrepancy with free Z3 performance should be discussed in more detail, especially regarding spectral mismatch and structural constraints.
  • The morphological transformation from fibrous aggregates to spiky nanoparticles is compelling. Still, the reproducibility of this transformation across batches should be addressed.
  • The UV-Vis and FTIR data suggest strong interactions between CDP and Z3, but the manuscript lacks complementary data such as NMR or Raman spectroscopy to confirm the nature of these interactions.
  • The SEM and TEM images are informative, but quantitative analysis of particle size distribution (e.g., using ImageJ or statistical plots) would enhance the rigor of the morphological characterization.
  • The cytotoxicity and phototoxicity assays are well-executed; however, the manuscript should include data on long-term cell viability and apoptosis pathways to strengthen the therapeutic claims.
  • The use of HeLa cells is appropriate, but testing on additional cancer cell lines (e.g., MCF-7, A549) would improve the generalizability of the findings.
  • The manuscript lacks in vivo data, which is critical for validating the clinical relevance of the proposed photothermal agent.
  • The discussion section is rich in interpretation but would benefit from a more critical comparison with benchmark agents such as ICG or commercial Pt-based nanoparticles.
  • The term “urchin-like” is used to describe morphology; consider using standardized terminology or referencing similar structures in literature for clarity.
  • The synthesis method involving glutaraldehyde crosslinking should include a discussion on potential cytotoxicity or residual aldehyde content.
  • The manuscript does not address the stability of the nanoparticles in physiological conditions (e.g., serum, pH variation), which is essential for biomedical applications.
  • The photothermal cycling stability is promising, but the manuscript should include degradation or aggregation studies post-irradiation.
  • The figures are generally well-presented, but some lack scale bars or resolution details. Ensure all figures are standardized and labeled consistently.
  • The manuscript would benefit from a graphical abstract summarizing the synthesis, structure, and therapeutic mechanism of the CDPNCs-Z3 nanoparticles.

Reviewer 3 Report

Comments and Suggestions for Authors

The paper Self-Assembled Nanoparticles from Cationic Dipeptides and D-π-A Chromophores for Near-Infrared Photothermal Therapy reports on the design of novel supramolecular and biocompatible CDPNCs-Z3 composite nanoparticles characterized by high photothermal performance, cycling stability, high cellular penetration, negligible dark toxicity, and potent photothermal killing effect. The study combines a number of investigation techniques including SEM and TEM analysis, UV/VIS and FTIR spectroscopies as well as assessments of photothermal performance and internalization of the composite in HeLa cells. The paper objective is clearly stated, Introduction is concise and relevant and Figures are of excellent quality and very informative. However, the interpretation of some results is questionable and the in-depth discussion of obtained results is lacking. After a major revision, I suggest the publication of this paper in International Journal of Molecular Sciences. Below is the list of questions/comments that should be addressed to Authors.

1. Results and Discussion:

a) The caption of the Figure 5a should be revised – divided to a) (UV/VIS) and b) (FTIR) part. The UV/VIS part needs to be supplemented with specified solvent, exact concentration of each sample and temperature.

b) The major issue regarding UV/VIS method is the selection of DMSO as a solvent, especially when the prepared sample is diluted and injected to intracellular aqueous solution. In general, the solvent effect is known to have a high impact on electronic structure. Why did Authors use DMSO - please elaborate.

c) Why Authors didn’t obtain UV/VIS spectra by preparing diluted aqueous solutions or solid UV/VIS spectra of samples by using Diffuse-reflectance spectroscopy?

d) The results of UV/VIS-spectroscopy are superficially and tentatively interpreted – the clearly visible shoulders around 530 nm and 670 nm, appeared upon self-assembly, are neither mentioned neither discussed – please elaborate?                                                                  Furthermore, which reliable and straightforward evidences did Authors provide in order to introduce pi-pi interactions and a possible ET to explain enhancement in band intensity. ET would result in detectable charge-separation, which could be evidenced with EPR while pi-pi interactions can be elucidated from known crystal structure.

e) Regarding FTIR, it is known that when working with supramolecular systems, the applied pressure can in some cases lead to substantial structural perturbations leading to different electronic structure and optical properties. Why did Authors use pellets instead of ATR-FTIR which allows one to obtain the spectra of as-prepared pure sample?

f) the first two paragraphs of Discussion section are well suited for Conclusion and not for discussion of the obtained results. Consider revising.

g) The Discussion section should incorporate interpretation and discussion of results emphasizing already known results and discussions of similar or comparable systems from literature sources. This is completely missing. Please elaborate.

2. Materials and Methods

a) Subsection 4.1. is lacking important details about the synthetic procedure (e.g. quote An appropriate amount of Z3 was then added….,) and the dilution process. Please comment.

Did Authors use the optimal CDP: Z3 mass ratio of 1:2?

b) Regarding the purity of the dark-green precipitate - how are Authors sure that washing the sample until the supernatant became nearly colorless is good enough to get a sample of a high purity? Are there any experimental techniques that could be used to check the purity, such as thin-layer chromatography? Washing liquid should be also specified.

3. Conclusion

The first two paragraphs are well suited for Introduction and not for Conclusion. Consider revising.

Round 2

Reviewer 1 Report

Comments and Suggestions for Authors

Dear Authors. 
On page six, the sentence that says: "... and SEM histogram in Figure 3f..." This does not seem correct.  Could you verify it, please.  As soon as it makes this correction, the manuscript will be accepted.

Author Response

Response to Reviewer 1 comments

Manuscript Number: ijms-3966673

Dear reviewer,

Thank you for your thoughtful reading and kind feedback. Your comments have helped us to significantly clarify and strengthen the presentation of our work.

Comment:On page six, the sentence that says: "... and SEM histogram in Figure 3f..." This does not seem correct.  Could you verify it, please. As soon as it makes this correction, the manuscript will be accepted.

Response: We are grateful to you for pointing out this discrepancy. The sentence on page six has been revised to accurately reference the SEM histogram, which is now correctly cited as Figure 2f. We apologize for this oversight and thank you for helping us improve the manuscript’s accuracy.

Reviewer 2 Report

Comments and Suggestions for Authors

The author has made all the required basic revisions, and I believe the paper can be accepted after ensuring that all reviewers' comments have been addressed.

Author Response

Response to Reviewer 2 comments

Manuscript Number: ijms-3966673

Dear reviewer,

Thank you for your thoughtful reading and kind feedback. Your comments have helped us to significantly clarify and strengthen the presentation of our work.

Comment:The author has made all the required basic revisions, and I believe the paper can be accepted after ensuring that all reviewers' comments have been addressed.

Response: We thank the reviewer for their positive assessment and recommendation. We confirm that we have carefully addressed all comments from all reviewers, and we believe the manuscript is now greatly improved and ready for publication.

Reviewer 3 Report

Comments and Suggestions for Authors

I wish to thanks the Authors for accepting a majority of suggestions and for providing argumented answers and comments.

Author Response

Response to Reviewer 3 comments

Manuscript Number: ijms-3966673

Dear reviewer,

Thank you for your thoughtful reading and kind feedback. Your comments have helped us to significantly clarify and strengthen the presentation of our work.

Comment:I wish to thanks the Authors for accepting a majority of suggestions and for providing argumented answers and comments.

Response: We thank you for your positive feedback and for acknowledging our efforts to address your suggestions. We are grateful for your constructive comments, which have helped us to improve the manuscript.